# Metabolic and Transcriptional Stress Memory in *Sorbus pohuashanensis* Suspension Cells Induced by Yeast Extract

**DOI:** 10.3390/cells11233757

**Published:** 2022-11-24

**Authors:** Yuan Li, Zhi-Qiang Luo, Jie Yuan, Sheng Wang, Juan Liu, Ping Su, Jun-Hui Zhou, Xiang Li, Jian Yang, Lan-Ping Guo

**Affiliations:** 1State Key Laboratory Breeding Base of Dao-di Herbs, National Resource Center for Chinese Materia Medica, China Academy of Chinese Medical Sciences, Beijng 100700, China; 2School of Traditional Chinese Medicine, Guangdong Pharmaceutical University, Guangzhou 510006, China

**Keywords:** *Sorbus pohuashanensis*, suspension cells, yeast extract, stress memory, biphenyl phytoalexin

## Abstract

Plant stress memory can provide the benefits of enhanced protection against additional stress exposure. Here, we aimed to explore the responses of recurrent and non-recurrent yeast extract (YE) stresses in *Sorbus pohuashanensis* suspension cells (SPSCs) at metabolomics and transcriptional levels. Biochemical analyses showed that the cell wall integrity and antioxidation capacity of SPSCs in the pretreated group were evidently improved. Metabolic analysis showed that there were 39 significantly altered metabolites in the pretreated group compared to the non-pretreated group. Based on the transcriptome analysis, 219 differentially expressed genes were obtained, which were highly enriched in plant–pathogen interaction, circadian rhythm–plant, oxidative phosphorylation, and phenylpropanoid biosynthesis. Furthermore, the correlation analysis of the transcriptome and metabolome data revealed that phenylpropanoid biosynthesis involved in the production of biphenyl phytoalexins may play a critical role in the memory response of SPSC to YE, and the key memory genes were also identified, including *PAL1*, *BIS1*, and *BIS3*. Collectively, the above results demonstrated that the memory responses of SPSC to YE were significant in almost all levels, which would be helpful for better understanding the adaptation mechanisms of medicinal plants in response to biotic stress, and laid a biotechnological foundation to accumulate favorable antimicrobial drug candidates from plant suspension cells.

## 1. Introduction

In nature, when plants encounter adverse environmental conditions such as extreme temperatures, excessive incident light, drought, flooding, and salinity, as well as pathogen infection and herbivore infestation, they may produce some alarm signals and exhibit sophisticated protection mechanisms to overcome stress factors and acclimatize to the new situation [1,2]. Although most of the altered stress responses would be recovered to the basal level after stress release, plants could efficiently integrate these environmental signals into a “stress memory”, which subsequently prepare themselves for future stresses [3,4]. It is thought that “stress memory” is programmed by chemical modifications such as methylation to DNA known as epigenic markers. There seem to be long- and short-term memories, which may be distinguished by their stability in the same generation or hereditary to the future generation [3,5]. In recent years, the mechanisms involved in the processes of plant stress memory have been increasingly described, ranging from epigenetic modification and the regulation of gene and protein expression to the transient accumulation of metabolites [6]. Notably, a majority of the 200,000 known secondary metabolites in plants have been reported to participate in the plant’s responses to the stress environment [2,7,8], such as arabidopsides in two *Arabidopsis* accessions (Col-0 and N14) [9], abscisic acid in rice [10], and anthocyanins in wheat [11]. Therefore, studying the changes in the abundance of metabolites and the related metabolic regulatory networks could be key to clarifying the molecular basis of plant stress memory.

*Sorbus pohuashanensis (SP)* is a member of the family Rosaceae (subfamily Maloideae) [12], whose fruits are edible and can be made into juices, jams, and pies [13]. It can also be used as an herbal medicine for relieving cough, resolving phlegm, and strengthening the spleen, according to traditional Chinese medicine (TCM) theory [12], which contains a variety of chemical constituents, mostly terpenes, flavonoids, and organic acids [14,15]. Similar to other fruit trees, pathogen infection can cause huge economic damage to SP [16]. *Sorbus pohuashanensis* suspension cells (SPSCs) is a good and robust in vitro model for rapid accumulation and mechanism exploration of the secondary metabolites produced by plants under stress conditions [15]. In our previous work, we found that the treatment of SPSCs with yeast extract (YE, a fungal elicitor) rapidly induced the formation and accumulation of biphenyl phytoalexins, a class of secondary metabolites that could be produced in plants under both biotic and abiotic stress conditions [14,17]. We further explored the antimicrobial activities of these phytoalexins and found that they could significantly inhibit the growth of pathogenic fungi and drug-resistant bacteria, such as *Alternaria tenuissi*, methicillin-resistant *Staphylococcus aureus*, and *Physalospora piricola*, which have broad application prospects in the fields of natural pesticides and drug discovery [14,16]. Interestingly, we found that several biphenyl phytoalexins were involved in the memory response of *Sorbus aucuparia* (the same genus of SP) suspension cells to YE [18]. However, the comprehensive metabolic memory response of SPSCs after treatment with YE and its underlying molecular regulatory mechanisms still remain unclear.

In this work, the cell wall integrity and antioxidation capacity of SPSCs induced by recurrent YE stress were firstly evaluated. Then, a non-targeted metabolomics analysis was conducted to evaluate the metabolomic memory responses of specific secondary metabolites in SPSCs. Next, RNA-seq analysis was performed to elaborate transcriptional memory responses and identify key genes and pathways to increase the YE resistance of SPSCs. Finally, a correlation analysis of metabolomic and transcriptional results was performed to comprehensively illustrate the memory responses of SPSCs to YE. Our study may provide insights into the metabolic and transcriptional memory responses as well as triggering processes of SPSCs to fungal elicitors, which would provide an efficient strategy for enhancing the production of desirable secondary metabolites in SPSCs.

## 2. Materials and Methods

### 2.1. Cell Culture, Treatment, and Experimental Design

The SPSCs were cultured in MS medium at room temperature and placed on a shaking table at 120 r min^−1^ under dark conditions, as described in our previous report [17]. Yeast extract (Oxoid, Basingstoke, Hampshire, UK; cat. no. LP0021B) was dissolved in distilled water and sterilized before being added to the cell cultures at the final concentration of 3 g L^−1^. Then, 5-day-old SPSCs were divided into four groups, including PT_T (pretreated group), PT_NT (resumed group), NT_T (non-pretreated group), and NT_NT (control) groups (Figure 1). SPSCs in PT_T group were subjected to repeated YE stresses as follows: 5-day-old SPSCs were pretreated (PT) with YE for 12 h, and then the culture medium was removed and the fresh medium was added. After being sub-cultured for 5 days, the SPSCs were treated (T) with YE. SPSCs in the PT_NT group were subjected to a YE stress as follows: 5-day-old SPSCs were pretreated with YE for 12 h, and then the culture medium was removed and the fresh medium was added. After being sub-cultured for 5 days, the SPSCs were non-treated (NT) with YE, using sterile distilled water instead. The SPSCs in the NT_T group were subjected to a YE stress as follows: 5-day-old SPSCs were treated with sterile distilled water for 12 h, and then the culture medium was removed and the fresh medium was added. After being sub-cultured for 5 days, the SPSCs were treated with YE. The SPSCs in the NT_NT (control) group were non-treated with YE and incubated under normal culture conditions throughout the whole experimental procedure. The cells were collected at 0, 6, 12, 24, and 48 h after treatment (HAT) for further analysis.

### 2.2. Determination of Cellulose, Oxidative Stress Parameters, and Antioxidant Enzyme Levels 

The content of cellulose was measured using a commercial assay kit (Solarbio Ltd., Beijing, China, cat. no. BC4280) based on the anthrone reagent method, as described by Updegraff [19]. The production rate of superoxide anions (O_2_^•−^) was determined using a commercial assay kit (Solarbio Ltd. Beijing, China, cat. no. BC1295) based on the method of hydroxylamine oxidation, as described by Bors et al. [20]. The content of hydrogen peroxide (H_2_O_2_) was determined using a commercial assay kit (Solarbio Ltd. Beijing, China, cat. no. BC3595) based on the titanium sulfate method, as described by Satterfield and Bonnell [21]. Catalase (CAT) activity was quantified by measuring the consumption of H_2_O_2_ at 240 nm [22] using a commercial assay kit (Solarbio Ltd. Beijing, China, cat. no. BC0205). Superoxide dismutase (SOD) activity was assayed by measuring its ability to inhibit the formation of the blue formazan (560 nm) [23] using a commercial assay kit (Solarbio Ltd. Beijing, China, cat. no. BC0175). Peroxidase (POD) activity was determined by guaiacol colorimetry (470 nm) [24] using a commercial assay kit (Solarbio Ltd. Beijing, China, cat. no. BC0095). The above assays were performed following the manufacturer’s instructions. 

### 2.3. Metabolite Extraction, UPLC-QTOF/MS Analysis, and Data Processing

The obtained SPSCs were dried at 40 °C and ground to a fine powder. Then, the cell powder (50 mg) was dissolved in methanol (1.5 mL) under ultrasonic extraction for 30 min. After centrifugation (12,000 rpm, 10 min), the supernatant was filtered through a 0.22 μm microporous membrane, and the filtrate was used for metabolome analysis.

The prepared samples were analyzed by ultrahigh performance liquid chromatography–quadrupole time-of-flight mass spectrometry (UPLC-QTOF/MS) (Waters, Milford, MA, USA). The UPLC conditions were as follows: Waters ACQUITY UPLC BEH C_18_ (2.1 mm × 100 mm, 1.7 µm) column; temperature, 40 °C; solvent system, 0.1% formic acid–water (A), 0.1% formic acid–acetonitrile (B); flow rate, 0.50 mL min^−1^; injection volume, 1 µL; gradient program: 0–1 min, 95% A; 1–6 min, 95% → 60% A; 6–8 min, 60% → 48% A; 8–12 min, 48% → 15% A; 12–15 min, 15% → 2% A, respectively. The MS conditions were as follows: negative ion mode; scanning range, 50–1500 Da; capillary voltage, 2500 V; sampling cone voltage, 35.0 V; offset tube voltage, 80.0 V; desolvation temperature, 500 °C; cone airflow, 50 L h^−1^; desolvation airflow, 800 L h^−1^. Correction solution: Leucine enkephalin (*m*/*z* = 554.2615).

The collected raw data, including sample information, ion intensities, *m*/*z* values, and retention times, was imported into Progenesis QI software (https://www.nonlinear.com/progenesis/qi/, accessed on 29 May 2022) for peak alignment, peak picking, and peak annotation. The processed data were then imported into SIMCA 14.1 for OPLS-DA and VIP analysis to maximize the separation of the groups and screen differential metabolites. The metabolites were identified or tentatively characterized by comparing with the data from the published literature [14,15] and chemical databases, such as PubChem (https://pubchem.ncbi.nlm.nih.gov/, accessed on 29 May 2022), Chemspider (http://www.chemspider.com/, accessed on 29 May 2022), KNApSAcK (http://www.knapsackfamily.com/KNApSAcK/, accessed on 29 May 2022), and Metlin (http://metlin.scripps.edu, accessed on 29 May 2022).

### 2.4. RNA Extraction, Library Preparation, and Sequencing

The total RNA was extracted from SPSCs by TRIzol reagent (Thermo Fisher Scientific, Waltham, MA, USA) according the manufacturer’s protocol. The total RNA concentration was determined by a NanoDrop 2000 spectrophotometer (Thermo Fisher Scientific, Waltham, MA, United States), and its quality was evaluated by an Agilent 2100 bioanalyzer. The RNA-seq library was built with the mRNA-seq Sample Preparation Kit (Illumina, San Diego, CA, USA) following the manufacturer’s protocol and then sequenced on an Illumina HiSeq^TM^ 2500 platform. The transcriptome sequencing and analysis were conducted by Nuohe Zhiyuan Technology Co., Ltd. (Beijing, China). Differentially expressed genes (DEGs) were identified by Package EdgeR (*q* value < 0.05, |log2(Fold Change, FC) | ≤ 2) [25] and annotated to NCBI non-redundant protein sequences (NR), nucleotide sequences (NT), Kyoto Encyclopedia of Genes and Genomes (KEGG), a manually annotated and reviewed protein sequence database (SwissProt), protein family (PFAM) database, Gene Ontology (GO), and euKaryotic Ortholog Groups (KOG) database. Then, functional and pathway enrichment analyses of DEGs were performed using GOSeq R package [26] and KOBAS software (http://www.genome.jp/kegg/, accessed on 29 May 2022) [27], respectively.

### 2.5. Statistical Analysis

All experiments were performed with at least three biological replicates and the data were presented as mean ± standard deviation (SD). Statistical significance was analyzed by one-way ANOVA, and the means were compared by Tukey’s test (*p* < 0.05).

## 3. Results

### 3.1. Cell Wall Integrity of SPSCs Induced by Recurrent YE Stress

The plant cell wall, a formidable and dynamic barrier, acts as the hub of stress surveillance and response [28]. Cellulose is the major component of the plant cell wall, and its degradation and/or the inhibition of its biosynthesis can be generally triggered by microbial attack, leading to the degradation of the cell wall [29,30]. In this work, although no significant differences of the cellulose content were observed between PT_T and PT_NT treatments at 0, 6, 12, and 24 HAT, the content of cellulose in the PT_T treatment was evidently higher than that in the PT_NT treatment at 48 HAT (Figure 2A). The increased cellulose production could contribute to the maintenance of the cell wall integrity, which might reflect the ability of SPSCs to self-protect against YE, which was improved after repeated stress. 

### 3.2. Antioxidation Capacity of SPSCs Induced by Recurrent YE Stress

Under extreme stress conditions, the excessive production of reactive oxygen species (ROS), including oxygen radicals (e.g., superoxide anion) and certain non-radicals (e.g., hydrogen peroxide), are frequently observed in plants, leading to progressive oxidative damage and ultimately cell death [31,32]. To combat oxidative stress, plants have evolved an efficient antioxidant system, especially ROS-scavenging enzymes such as CAT, SOD, and POD [33,34]. In this study (Figure 2B–F), although no significant changes of O_2_^•−^ and H_2_O_2_ contents were found in the NT_NT, NT_T, and PT_T groups after YE treatment, higher levels of CAT (at 6 and 24 HAT), SOD (all time points), and POD (at 0, 6, 12, and 24 HAT) were observed in the PT_T treatment compared with the NT_T treatment. Interestingly, the activity of POD was increased in PT_T treatment at 0–6 HAT, while a slightly decreased POD activity was observed in the NT_T treatment at the same time. These results indicated that the memory response was evident in SPSCs to YE, which could contribute to alleviating oxidative damage during recurrent YE stress. In addition, the significantly higher SOD and POD activities observed in the PT_T and PT_NT treatments at 0 HAT may be due to the retention of stress memory for a certain time.

### 3.3. Metabolomic Memory Response in SPSCs Induced by YE

Plants biosynthesize a variety of natural products, such as defense-related phytohormones and other metabolites, to combat the adverse impact of environmental stresses [35]. To explore the role of the accumulation of specific secondary metabolites in the memory response of SPSCs to YE, a non-targeted metabolomics analysis was performed using the UPLC-QTOF/MS system. In total, 4395 precursor ions were extracted with Progenesis QI software (Appendix A) across all samples (*p* < 0.01), from which 39 metabolites, including nine biphenyls, nine triterpenes, seven organic acids, three amino acids, eight nucleotides, and three carbohydrates, were identified/annotated (Appendix A). 

Compared with NT_NT, the differentially expressed metabolites (DEMs) in NT_T and PT_T at 6, 12, 24, and 48 HAT were obtained, respectively. Then, a Venn diagram analysis was performed to characterize the overlap of DEMs between NT_T and PT_T (Figure 3A). Obviously, more up-accumulated metabolites in PT_T than those in NT_T at all time points were analyzed. However, for the down-accumulated metabolites, the non-pretreated SPSCs had more metabolites than the pretreated SPSCs.

We further compared the metabolic features of the pretreated SPSCs with those of the non-pretreated SPSCs. The OPLS-DA analysis of the resulting metabolite data indicated a clear separation between the NT_T and PT_T groups, which suggests that repeated YE stress affected the metabolite accumulation of SPSC to a certain extent (Appendix A). Then, four heatmaps (Figure 3B–E) were generated as graphical representations of the expression changes of the metabolites (VIP > 1) between the NT_T and PT_T groups at 6, 12, 24, and 48 HAT. Overall, the expressions of most metabolites in the PT_T group were higher than those in the NT_T group at each time point analyzed. An in-depth comparative analysis revealed that most metabolites related to the resistance to pathogens, especially biphenyls (including 2′-hydroxyaucuparin, 2′-hydroxyaucuparin-2′-*O*-glu, 2′-hydroxyaucuparin-*O*-di-glu, noraucuparin, noraucuparin-5-*O*-*β*-d-glu, noraucuparin-4-*O*-*β*-d-glu and aucuparin, sorbusins A, and sorbusins B), were up-accumulated in the PT_T group at 6, 12, 24, and 48 HAT (Figure 4). At 0 HAT, most of the biphenyls were undetectable in all groups, which was consistent with our previous findings [17]. Interestingly, an extremely low content of 2′-hydroxyaucuparin and hydroxyaucuparin-*O*-di-glu could be detected in PT_T at 0 HAT and PT_NT at 0, 6, 12, 24, and/or 48 HAT, which may be due to the retention of stress memory for a certain time [36].

Collectively, the above results demonstrated that stress memory could be developed in SPSCs under YE stress, and the accumulation of the biphenyls in SPSCs might reflect a stress memory response to YE for copying with pathogen infection. In addition, our results may contribute to the identification and accumulation of SPSC-derived metabolites as potential therapeutic agents.

### 3.4. Transcriptional Memory Response in SPSCs Induced by YE

Plants resort to an extensive transcriptional reprogramming to overcome biotic and abiotic stress constraints [37]. To explore the transcriptional memory types of SPSCs in response to YE, RNA-seq analysis has been performed to reveal the stress-related genes and pathways that potentially exhibit a memory behavior. Compared with the NT_NT group, the number of differentially expressed genes (DEGs) in the NT_T and PT_T groups at 0, 6, 12, 24, and 48 HAT were obtained, as shown in Figure 5A,B. Notably, most DEGs were observed at 6 HAT in both groups, suggesting that the transcription stress responses may be more significant at 6 HAT than the other time points.

Compared with NT_NT, 9703 upregulated and 9389 downregulated genes (YE-responsive transcripts, YERT) were found in the NT_T treatment at 6 HAT (Figure 5A). Then, GO analysis was performed to understand the biological functions of YERT, including biological process (BP), cell component (CC), and molecular function (MF). As shown in Figure 5C, the top five enriched BP ontologies were dominated by protein phosphorylation, phosphorylation, phosphorus metabolic process, phosphate-containing compound metabolic process, and oxidation-reduction process. The top five enriched MF ontologies were dominated by protein kinase activity, kinase activity, oxidoreductase activity (acting on paired donors, with the incorporation or reduction of molecular oxygen), phosphotransferase activity (alcohol group as acceptor), and transferase activity. Furthermore, the cell wall, external encapsulating structure, mitochondrial inner membrane presequence translocase complex, microtubule, and apoplast were ranked as the top five CC ontologies. The GO analysis indicated that YE could stimulate the stress-responsive genes to induce cell wall and membrane damage as well as oxidative stress in SPSCs, which was consistent with the biochemical findings.

Next, KEGG pathway enrichment analysis was performed and the top five enriched pathways are shown in Table 1, including phenylpropanoid biosynthesis, terpenoid backbone biosynthesis, plant–pathogen interaction, flavonoid biosynthesis, and plant hormone signal transduction (*p* < 0.05). The result indicated that the stress-responsive genes in SPSCs would be activated to mainly produce various secondary metabolites (such as phenylpropanoids, terpenoids, and flavonoids) to resist YE stress, which was consistent with the metabolomic findings.

Compared with NT_T, 206 upregulated and 13 downregulated genes (potential memory-changed transcripts, PMCT) were found in the PT_T treatment at 6 HAT (Appendix A). Then, KEGG pathway enrichment analysis was performed and the significant pathways are shown in Table 1, including plant–pathogen interaction, circadian rhythm–plant, oxidative phosphorylation, and phenylpropanoid biosynthesis (*p* < 0.05). Interestingly, both YERT and PMCT were highly enriched in plant–pathogen interaction and the phenylpropanoid biosynthesis pathway, suggesting that the transcriptional memory of the genes involved in promoting phenylpropanoid biosynthesis may enhance the stress tolerance of SPSCs in response to YE, which was in agreement with the differentially expressed metabolites (biphenyl phytoalexins) as revealed by the metabolomics analysis. 

### 3.5. Correlation Analysis between Transcripts and Phytoalexins 

To reveal the candidate genes in the regulation of phenylpropanoid biosynthesis in SPSCs induced by YE and further identify the memory genes, correlation analysis between the DEGs and biphenyl phytoalexins was carried out. By comparison with NT_NT, the transcript levels of 114 genes in the phenylpropanoid biosynthesis pathway was highly correlated with the accumulations of the nine phytoalexins in NT_T treatment at 6 HAT, including 5 *4CLs*, 13 *BGLs*, 4 *CCRs*, 5 *CADs*, 1 *CSE*, 4 *CYP73As*, 1 *CYP84A*, 1 *CYP98A*, 6 *HCTs*, 24 *OMTs*, 15 *POXs*, 1 *REF1*, 5 *BISs*, 6 *B4Hs*, 1 *PAL*, and 22 *UGTs* (Appendix A). Furthermore, by comparison with NT-T, 25 DEGs encoding phytoalexin biosynthesis enzymes in PT_T treatment at 6 HAT were found to be associated with stress memory response, including 1 *PAL*, 2 *BISs*, 11 *OMTs*, 5 *B4Hs* and 6 *UGTs* (Appendix A). Among them, *PAL* (phenylalanine ammonia lyase) is a key enzyme in phenylpropanoid metabolism [38], and *BIS* (biphenyl synthase) has been proven to play a critical role in the biosynthesis of the biphenyl skeleton [39,40]. Thus, *PAL1* (Cluster—19734.30063), *BIS1* (Cluster—19734.34421), and *BIS3* (Cluster—19734.13585) would be the key memory genes of SPSCs in response to YE.

The time course of expression of key memory genes in YE-treated SPSCs was also determined, as shown in Figure 6. Obviously, the three genes showed considerably higher transcript levels in the PT_T group than those in the NT_T group after YE treatment. Moreover, the transcript levels of *PAL1*, *BIS1*, and *BIS3* and were increased and peaked at 12, 6, and 6 HAT, respectively, and then they decreased. These results further demonstrated the importance of the three genes in the regulation of the memory responses of SPSCs to YE, and also suggested that transcriptional memory would gradually fade over time if the stress stimulation signal was not strengthened.

## 4. Discussion

YE has been recognized as an effective biotic elicitor owing to its high content of glucan, chitin, ergosterol, vitamin B complex, and glycopeptides, as these components could elicit plant defensive responses by triggering enhanced phytoalexin accumulation [41,42]. It has been noted that YE elicitation drastically increased biphenyl phytoalexin yield in *Sorbus aucuparia* (the same genus of SP) cell cultures, while other elicitors, such as *Erwinia amylovora*, *Venturia inaequalis*, and methyl jasmonate, did not show their effectiveness in the production of biphenyl phytoalexins [43]. In our previous studies, we also found that the effects of YE on biphenyl phytoalexin accumulation were more pronounced than other elicitors and these compounds showed evident antimicrobial activities against pathogenic fungi and drug-resistant bacteria [14,16,17]. That is why we selected YE as the elicitor in this work. Since plants usually produce a common set of signals and active substances to defend themselves from different types of stresses [44], the combination of YE with other elicitors such as salicylic acid, sorbitol, and heavy metals may be more effective for the enhancement of biphenyl phytoalexin production in SPSCs.

A plant previously challenged by a stress event might improve its future stress responses, becoming more tolerant and/or resistant via a memory acquisition process [45]. For example, Szechyńska-Hebda et al. indicated that plants possess a complex and dynamic light training and memory system for optimizing light acclimation and immune defenses [46]. In addition, photosystem II 22kDa protein was further proven to be an important regulator of chloroplast retrograde signaling for light memory [47]. While the mechanism of the general stress response has become better understood, much remains to be uncovered about the role of stress memory. Based on our previous studies [18], we hypothesized that SPSCs could exhibit YE stress memory when recurrent YE stress was imposed. The elucidation of its underlying mechanisms would contribute to the production of bioactive compounds, especially biphenyl phytoalexins.

The plant cell wall is the front line for sensing and responding to various external stimuli [48]. It is a complex polysaccharide network predominantly formed by a mixture of cellulose, pectin, and hemicellulose [49]. As the main structural component of the cell wall, cellulose consists of *β*-1, four linked glucan chains that join together to form microfibrils and interconnected by hemicelluloses (rich in xyloglucan) and pectins (rich in galacturonic acid) [50,51]. During a pathogen attack, the plant cell wall would be degraded or weakened by the microbe-derived compounds, leading to the generation of wall fragments, such as cellobiose, oligogalacturonides, and homogalacturonan-derived oligomers [52]. These fragments are able to be recognized by cell surface receptors such as wall-associated kinases (WAKs) and act as signaling molecules to induce wall-related gene expression for the modification of the wall [53]. Our study on YE-induced alterations in wall composition is limited and the wall composition-related genes need to be further explored to gain in-depth knowledge about the stress memory process.

In addition to the increased cellulose production, the activities of CAT, SOD, and POD in SPSCs were enhanced after repeated YE stress. Since they were the specific enzymes to counteract ROS-induced cell damage, their increase after YE elicitation suggested that the overproduction of ROS (O_2_^•−^ and H_2_O_2_) occurred in SPSCs. Accumulating evidence suggests that ROS are produced transiently by plant cells in response to pathogen infection, known as the ‘oxidative burst’ [54]. If we measured the contents of O_2_^•−^ and H_2_O_2_ at 0.5 HAT-1 HAT, significant differences of ROS production may be observed among the NT_NT, NT_T and PT_T groups, as described by Qiu et al. [55]. They also proposed that YE-induced H_2_O_2_ activated the *BIS* gene and the biphenyl biosynthetic pathway, which played an essential role in the accumulation of aucuparin [55]. In addition, research has indicated that stressed plants yield phenolic compounds to resist against oxidative stress [56]. Thus, the increased levels of biphenyls and other phenolics in SPSCs after single YE stress enhanced resistance, and this resistance was further strengthened by repeated YE stress.

The SPSCs pre-exposed to YE stress induced extensive transcriptional changes and the rapid accumulation of defense compounds, which has been remembered by SPSCs as stress memory and improved the stress response and tolerance under recurrent YE stress. Interestingly, the memory response of SPSCs induced by YE was highly correlated with the activation of *PAL1*, *BIS1*, and *BIS3*, which played essential roles in regulating biphenyl biosynthesis. Recently, epigenetic changes, especially the hypomethylation or hypermethylation of DNA, have been found to exert a profound influence on the stress-mediated transcriptional memory response [57]. Under stressful conditions, the overall increase in methylation contributes to maintaining genome stability, and demethylation of the resistance genes could enhance transcription to improve stress resistance [58]. Until now, there have been few reports regarding the regulatory relationship between DNA methylation and plant secondary metabolite accumulation during the stress memory process. Therefore, more research needs to be carried out to uncover the role of DNA methylation in regulating the expressions of biphenyl biosynthesis-related genes (especially *PAL1*, *BIS1*, and *BIS3*), which may provide further insights into the YE-induced stress memory in SPSCs.

## 5. Conclusions

In summary, the memory responses of SPSCs to YE were successfully unveiled and found to be evident by comprehensive analysis and comparison of the physiological changes, biochemical feedbacks, and the metabolomics and transcriptomics profiles. The results indicated that stress memory could contribute to alleviating cell wall and oxidative damage in SPSCs. After the correlation analysis between the metabolomic and transcriptional data, the phenylpropanoid biosynthesis pathway involved in the production of biphenyl phytoalexins was found to be mainly responsible for the stress memory responses. Three core genes, including *PAL1* (Cluster—19734.30063), *BIS1* (Cluster—19734.34421), and *BIS3* (Cluster—19734.13585), were highly associated with the biosynthesis of biphenyl skeleton and thus were considered as the key memory genes. The above results of this study may provide valuable information for developing proper priming techniques to accelerate the accumulation of more favorable secondary metabolites from plant suspension cells, which may be of benefit to drug or pesticide discovery and development.

## Figures and Tables

**Figure 1 cells-11-03757-f001:**
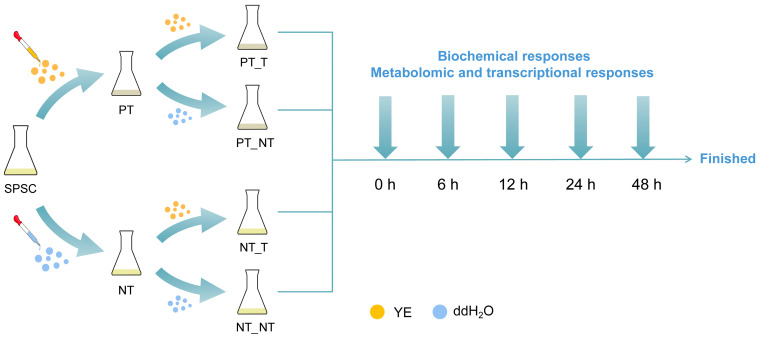
Schematic representation of the experimental design. SPSC, *Sorbus pohuashanensis* suspension cell; YE, yeast extract; PT, pretreated samples; NT, non-treated samples; T, treated samples; PT_T, pretreated group; PT_NT, resumed group; NT_T, non-pretreated group; NT_NT, control.

**Figure 2 cells-11-03757-f002:**
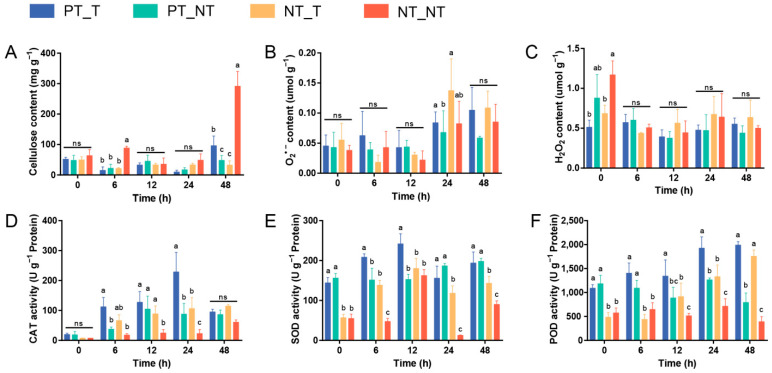
Biochemical analyses of cell wall integrity and antioxidation capacity of SPSC to YE stress. The levels of cellulose (**A**), O_2_^•−^ (**B**) and H_2_O_2_ (**C**), and the activities of CAT (**D**), SOD (**E**), and POD (**F**) in SPSCs. Different lowercase letters above columns indicate statistical differences at *p* < 0.05 by one-way ANOVA (*n* = 3). ns, not significant.

**Figure 3 cells-11-03757-f003:**
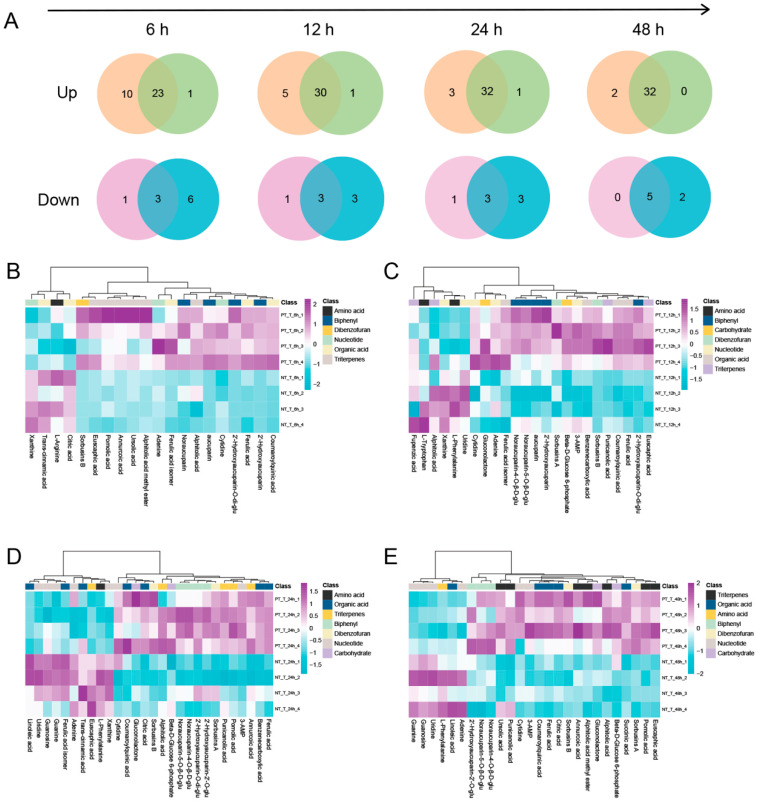
Global view of the distinct and common metabolites in SPSCs in response to YE. (**A**) Venn diagram showing the total number of the unique or overlapping metabolites between PT_T (**left**) and NT_T (**right**). Heatmaps of differential metabolites between PT_T and NT_T at 6 (**B**), 12 (**C**), 24 (**D**), and 48 (**E**) hours after treatment (HAT) with YE.

**Figure 4 cells-11-03757-f004:**
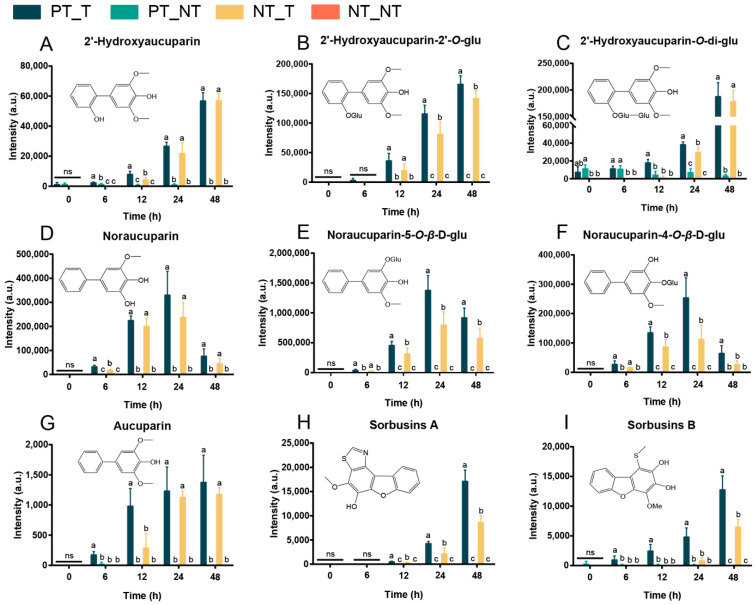
Contents of biphenyls in SPSCs in response to YE. (**A**) 2′-hydroxyaucuparin, (**B**) 2′-hydroxyaucuparin-2′-*O*-glu, (**C**) 2′-hydroxyaucuparin-*O*-di-glu, (**D**) noraucuparin, (**E**) noraucuparin-5-*O*-*β*-d-glu, (**F**) noraucuparin-4-*O*-*β*-d-glu, (**G**) aucuparin, (**H**) sorbusins A, and (**I**) sorbusins B. Different lowercase letters above columns indicate statistical differences at *p* < 0.05 by one-way ANOVA (*n* = 4). ns, not significant.

**Figure 5 cells-11-03757-f005:**
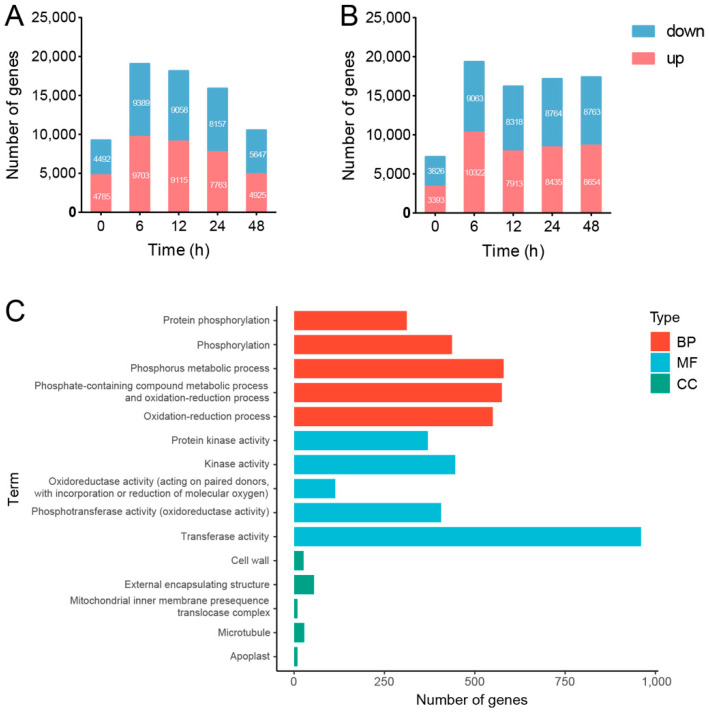
Global view of the differentially expressed genes (DEGs) in SPSCs in response to YE. (**A**) The number of DEGs between NT_T and NT_NT. (**B**) The number of DEGs between NT_T and PT_T. (**C**) GO term enrichment of DEGs between NT_T and NT_NT (TOP 5). BP, biological process; CC, cell component; MF, molecular function.

**Figure 6 cells-11-03757-f006:**
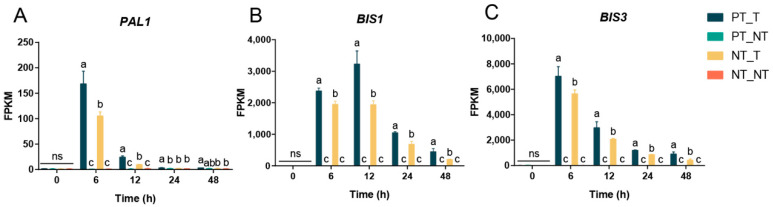
Expression levels of three selected genes in SPSCs in response to YE. (**A**) *PAL1*, (**B**) *BIS1*, and (**C**) *BIS3*. Different lowercase letters above columns indicate statistical differences at *p* < 0.05 by one-way ANOVA (*n* = 3). ns, not significant.

**Table 1 cells-11-03757-t001:** KEGG pathway enrichment of DEGs in NT_NT vs. NT_T and NT_T vs. PT_T.

KEGG Pathway	ID	*p*-Value (NT_NT vs. NT_T)	*p*-Value (NT_T vs. PT_T)
Phenylpropanoid biosynthesis ^a,b^	ko00940	2.72 × 10^−9^	0.045694574
Terpenoid backbone biosynthesis ^a^	ko00900	1.57 × 10^−5^	ns
Plant-pathogen interaction ^a,b^	ko04626	0.000256399	2.37 × 10^−9^
Flavonoid biosynthesis ^a^	ko00941	0.000422929	ns
Plant hormone signal transduction ^a^	ko04075	0.000495331	ns
Circadian rhythm–plant ^b^	ko04712	ns	0.011897832
Oxidative phosphorylation ^b^	ko00190	ns	0.025599648

^a^ Significant pathways in NT_NT vs. NT_T (Top 5); ^b^ significant pathways in NT_T vs. PT_T; ns, not significant.

## Data Availability

Data are available from corresponding author upon reasonable request.

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
