# Peer review of "Metabolic and Transcriptional Stress Memory in Sorbus pohuashanensis Suspension Cells Induced by Yeast Extract"

_cells, 2022, doi:10.3390/cells11233757_

Round 1
Reviewer 1 Report
In this manuscript, authors used combined transcriptomic and metabolomic analyses to investigate stress memory in Sorbus pohuashanensis suspension cells using various treatments with yeast extract as an elicitor. This approach helped to identify the possible involvement of phenylpropanoid biosynthesis in stress memory. The manuscript is generally well-written and I have just a few comments.
Line 14 defines YE in the abstract.
Consistency in using the abbr. SPSC for Sorbus pohuashanensis suspension cells e.g. in the introduction lines 49 and 51
The figures are generally well-prepared, however, I had difficulties reading them in the generated pdf. I suggest adjusting their size for better readability in the published manuscript.
Author Response
Response to Reviewer 1 Comments
Dear editors and reviewers,
Thank you for giving us this opportunity to submit a revised draft of the manuscript “Metabolic and transcriptional stress memory in Sorbus pohuashanensis suspension cells induced by yeast extract” (ID:1980630) for publication in Cells. We appreciate the time and effort that you and the reviewers dedicated to providing feedback on our manuscript and are grateful for the insightful comments on and valuable improvements to our paper. We have incorporated most of the suggestions made by the reviewers. Those changes have been highlighted in red in our revised manuscript. Please see below, in red, for a point-by-point response to the reviewers’ comments and concerns.
Referee #1:
In this manuscript, authors used combined transcriptomic and metabolomic analyses to investigate stress memory in Sorbus pohuashanensis suspension cells using various treatments with yeast extract as an elicitor. This approach helped to identify the possible involvement of phenylpropanoid biosynthesis in stress memory. The manuscript is generally well-written and I have just a few comments.
Response: We appreciate the positive and constructive feedback. We have provided detailed responses to the comments below.
Comments 1: Line 14 defines YE in the abstract.
Response: Thank you for pointing out this issue. The full name of the abbreviation of “YE” has been added to Abstract.
Comments 2: Consistency in using the abbr. SPSC for Sorbus pohuashanensis suspension cells e.g. in the introduction lines 49 and 51
Response: Thank you for pointing out this issue. We have checked the manuscript throughout and used the abbr. SPSCs for Sorbus pohuashanensis suspension cells.
Comments 3: The figures are generally well-prepared, however, I had difficulties reading them in the generated pdf. I suggest adjusting their size for better readability in the published manuscript.
Response: Thanks for your kind suggestion. We have adjusted the figure size in the manuscript and they become more readable.

Reviewer 2 Report
Review
The presented ms is devoted to several aspects of plant memory related to abiotic stresses. The Authors investigated changes in biphenyl phytoalexins, gene expression, and radicals due to YE stress and found that plant memory is related to phytoalexin and a set of genes. They also suggest that the cell wall is responsible for sensing stress.
In my opinion, the study is well-done. Still, some drawbacks need to be eliminated.
1. Please, state your working hypothesis,
2. Aims and ways they are to be solved should be modified to reflect the hypothesis,
3. It is not clear whether ANOVA requirements were verified. No information on ANOVA is given in the R section (or I have missed it...) 4. I prefer to separate the R and D sections as they allow independent data interpretation. Maybe it is worth separating R and D,
4. It would be interesting (at least in the D section) to discuss the role of b-glucans or pectins in parallel to cellulose as stress-sensing compounds and their role in plant memory in response to stresses.
After some revision, the ms could be published in the Cells journal.
Kind Regards,
Author Response
Response to Reviewer 2 Comments
Dear editors and reviewers,
Thank you for giving us this opportunity to submit a revised draft of the manuscript “Metabolic and transcriptional stress memory in Sorbus pohuashanensis suspension cells induced by yeast extract” (ID:1980630) for publication in Cells. We appreciate the time and effort that you and the reviewers dedicated to providing feedback on our manuscript and are grateful for the insightful comments on and valuable improvements to our paper. We have incorporated most of the suggestions made by the reviewers. Those changes have been highlighted in red in our revised manuscript. Please see below, in red, for a point-by-point response to the reviewers’ comments and concerns.
Referee #2:
The presented ms is devoted to several aspects of plant memory related to abiotic stresses. The Authors investigated changes in biphenyl phytoalexins, gene expression, and radicals due to YE stress and found that plant memory is related to phytoalexin and a set of genes. They also suggest that the cell wall is responsible for sensing stress.
In my opinion, the study is well-done. Still, some drawbacks need to be eliminated.
Response: We appreciate the positive and constructive feedback. We have provided detailed responses to the comments below.
Comments 1: Please, state your working hypothesis.
Response: Thanks for your kind suggestion. Based on our previous studies (Chin J Chin Mat Med, 2021, 46(10), 2467-2473.), we hypothesized that SPSCs could exhibit YE stress memory, when recurrent YE stress was imposed. And elucidation of its underlying mechanisms would contribute to the production of bioactive compounds, especially biphenyl phytoalexins (Please see the details in the second paragraph of Discussion).
Comments 2: Aims and ways they are to be solved should be modified to reflect the hypothesis,
Response: Thanks for your kind suggestion. The aims and ways have been modified, please see the manuscript, especially the Discussion part.
Comments 3: It is not clear whether ANOVA requirements were verified. No information on ANOVA is given in the R section (or I have missed it...)
Response: Thank you for pointing out this issue. The information on one-way ANOVA were added in the legends of Figure 2, Figure 4 and Figure 6.
Comments 4: I prefer to separate the R and D sections as they allow independent data interpretation. Maybe it is worth separating R and D,
Response: Thanks for your kind suggestion. The R and D sections have been separated and we have added more information in the Discussion section.
Comments 5: It would be interesting (at least in the D section) to discuss the role of b-glucans or pectins in parallel to cellulose as stress-sensing compounds and their role in plant memory in response to stresses.
Response: Thanks for your kind suggestion. The role of b-glucans or pectins in parallel to cellulose as stress-sensing compounds and their role in plant memory in response to stresses have been discussed in the manuscript (Please see the third paragraph of Discussion).

Reviewer 3 Report
This is an interesting paper reporting that effects of yeast extract (YE) stress on Sorbus pohuashanensis suspension cells (SPSCs) at metabolomics and transcriptional levels. Since the plant S. pohuashanensis is used for Chinese medicine, the results potentially contribute to herbal medicinal studies. Before its publication, the authors are requested to consider the following points.
Although “plant stress memory” could be a major keyword of this paper, introduction for the term with scientific definition cannot be found in the manuscript. In my understanding, “plant stress memory” is not widely accepted but is referred as to that plants integrate environmental signals into “stress memory” which is transmitted to the immediate progeny (Wibowo et al. 2015). It is thought that “stress memory” is programmed by chemical modifications such as methylation to DNA known as epigenic markers. There seem to be long- and short-term memories which may be distinguished by their stability in the same generation or hereditary to the future generation.
In this study the authors appear to use the term “stress memory” without clear definition. Even there is no statement on DNA methylation or glycosylation which has been believed to be a chemical mechanism to memorize environmental stresses. Although the authors emphasize “stress memory” throughout the manuscript, I would categorize this study to a transcriptome and motabolome analysis of yeast extract-treated Sorbus pohuashanensis suspension cells. Since the results are new and informative, I feel that the paper could stand without “stress memory” issue (of course, “stress memory” discussion is welcome).
<Others>
1. Abstract
Add full name of the abbreviation of “YE, line 14).
2. Materials and Methods
The materials and Methods section needs to be improved.
2.1. Yeast extract
In the authors’ group studies, yeast extract (YE) is very essential to induce microbial stress. However, I cannot find its YE’s specification in the article. Foe most readership, YE is not a representative elicitor but an organic nutrient for bacterial cultures. It is important to be reminded that the contents vary over the products. I suggest the authors to describe more details about YE (product company, quality) as well as the condition of the YE stress (concentration).
2.2 Assays
The authors just mentioned that biochemical assays followed the manufacturer’s instructions of commercial assay kits (Solarbio, Beijing, China). This is too unkind for the readership. Without obtaining the instruction manuals, no one can justify the procedures. The authors are requested to describe, at least, their detection principles used in Materials and Methods.
Author Response
Response to Reviewer 3 Comments
Dear editors and reviewers,
Thank you for giving us this opportunity to submit a revised draft of the manuscript “Metabolic and transcriptional stress memory in Sorbus pohuashanensis suspension cells induced by yeast extract” (ID:1980630) for publication in Cells. We appreciate the time and effort that you and the reviewers dedicated to providing feedback on our manuscript and are grateful for the insightful comments on and valuable improvements to our paper. We have incorporated most of the suggestions made by the reviewers. Those changes have been highlighted in red in our revised manuscript. Please see below, in red, for a point-by-point response to the reviewers’ comments and concerns.
Referee #3:
This is an interesting paper reporting that effects of yeast extract (YE) stress on Sorbus pohuashanensis suspension cells (SPSCs) at metabolomics and transcriptional levels. Since the plant S. pohuashanensis is used for Chinese medicine, the results potentially contribute to herbal medicinal studies. Before its publication, the authors are requested to consider the following points.
Response: We appreciate the positive and constructive feedback. We have provided detailed responses to the comments below.
Although “plant stress memory” could be a major keyword of this paper, introduction for the term with scientific definition cannot be found in the manuscript. In my understanding, “plant stress memory” is not widely accepted but is referred as to that plants integrate environmental signals into “stress memory” which is transmitted to the immediate progeny (Wibowo et al. 2015). It is thought that “stress memory” is programmed by chemical modifications such as methylation to DNA known as epigenic markers. There seem to be long- and short-term memories which may be distinguished by their stability in the same generation or hereditary to the future generation.
Response: Thanks for your comments and suggestions. The stress memory has been defined as genetic, epigenetic and physiological alterations that occur in response to stress, which subsequently prepare the plant for reoccurring stresses within or across generation(s). This scientific definition was recommended by Muhittin (Industrial Crops & Products, 2020, 154, 112695) and Wang, et.al.(Frontiers in Plant Science, 2016, 7, 501), which has been added in the Introduction part. And we have discussed the role of DNA methylation in the stress-mediated transcriptional memory response (Please see the last paragraph of Discussion).
In this study the authors appear to use the term “stress memory” without clear definition. Even there is no statement on DNA methylation or glycosylation which has been believed to be a chemical mechanism to memorize environmental stresses. Although the authors emphasize “stress memory” throughout the manuscript, I would categorize this study to a transcriptome and motabolome analysis of yeast extract-treated Sorbus pohuashanensis suspension cells. Since the results are new and informative, I feel that the paper could stand without “stress memory” issue (of course, “stress memory” discussion is welcome).
Response: Thanks for your comments and suggestions. The definition of the term “stress memory” has been added in the Introduction part and we have discussed the role of DNA methylation in the stress-mediated transcriptional memory response (Please see the last paragraph of Discussion). Stress memory-coordinated changes at the organismal, cellular, and various omics levels prepare plants to be more responsive to reoccurring stress within or across generation(s). With the development and deployment of various omics approaches (e.g., transcriptomics, epigenomics, degradomics, proteomics, and metabolomics) using high-throughput processing pipelines, a substantial volume of large-scale data has been generated and interrogated to link molecular function with agronomic performance, thus providing new insights into the mechanistic basis of plant stress memory (Trends in plant science, 2022, 27(7), 699-716). In this work, the responses of recurrent and non-recurrent yeast extract (YE) stresses in Sorbus pohuashanensis suspension cells (SPSCs) were explored at metabolomics and transcriptional levels. This should be included in plant stress memory.
<Others>
Comments 1: Abstract
Add full name of the abbreviation of “YE, line 14).
Response: Thank you for pointing out this issue. The full name of the abbreviation of “YE” has been added in the Abstract.
Comments 2: Materials and Methods
The materials and Methods section needs to be improved.
Response: Thanks for your kind suggestions. We have modified the Material and Methods section in the manuscript.
Comments 3: Yeast extract
In the authors’ group studies, yeast extract (YE) is very essential to induce microbial stress. However, I cannot find its YE’s specification in the article. Foe most readership, YE is not a representative elicitor but an organic nutrient for bacterial cultures. It is important to be reminded that the contents vary over the products. I suggest the authors to describe more details about YE (product company, quality) as well as the condition of the YE stress (concentration).
Response: Thanks for your comments and suggestions. YE has been recognized as an effective biotic elicitor owing to its high content of glucan, chitin, ergosterol, vitamin B complex and glycopeptides, as these components could elicit plant defensive responses by triggering enhanced phytoalexin accumulation (In Vitro Cellular & Developmental Biology-Plant, 2022, 58, 615–629; Plant Cell, Tissue and Organ Culture (PCTOC), 2020, 141, 661–667). It has been noted that YE elicitation drastically increased biphenyl phytoalexin yield in Sorbus aucuparia (the same genus of SP) cell cultures, while other elicitors, such as Erwinia amylovora, Venturia inaequalis and methyl jasmonate, didn’t show their effectiveness in the production of biphenyl phytoalexins (Journal of agricultural and food chemistry, 2010, 58(22), 11977–11984). In our previous studies, we also found that the effects of YE on biphenyl phytoalexin accumulation were more pronounced than other elicitors and these compounds showed evident antimicrobial activities against pathogenic fungi and drug-resistant bacteria (Molecules, 2016, 21(9), 1180; Fitoterapia, 2021, 152, 104914; Chemistry & Biodiversity, 2021, 18(5), e2100079). That’s why we selected YE as the elicitor in this work. Since plants usually produce a common set of signals and active substances to defend themselves from different types of stresses (Biotechnology Applied Biochemistry, 2007, 46(4), 191-196.), the combination of YE with other elicitors like salicylic acid, sorbitol and heavy metals may be more effective for enhancement of biphenyl phytoalexin production in SPSCs. The above information has been added in Discussion (the first paragraph).
In addition, the product company and the catalog number of YE have been provided (See 2.1). Please enter the catalog number to search for the detail information of YE including the quality. And the preparation and final concentration of YE have also been added in 2.1.
Comments 4: Assays
The authors just mentioned that biochemical assays followed the manufacturer’s instructions of commercial assay kits (Solarbio, Beijing, China). This is too unkind for the readership. Without obtaining the instruction manuals, no one can justify the procedures. The authors are requested to describe, at least, their detection principles used in Materials and Methods.
Response: Thanks for your kind suggestions. The basic detection principles of these biochemical assays have been added (Please see section 2.2), and we also provided the literature relevant to these biochemical assays, which would be helpful for readers to see the details of the detection principles. And we also provided the catalog number of these assay kits, please enter the catalog number to search for the detail information, especially the instruction manuals.
